# Identification, Quantification, and Characterization of the Phenolic Fraction of *Brunfelsia grandiflora*: In Vitro Antioxidant Capacity

**DOI:** 10.3390/molecules27196510

**Published:** 2022-10-02

**Authors:** Raquel Mateos, Norma Ramos-Cevallos, Americo Castro-Luna, Mariella Ramos-Gonzalez, Zoyla-Mirella Clavo, Miguel Quispe-Solano, Luis Goya, José-Luis Rodríguez

**Affiliations:** 1Department of Metabolism and Nutrition, Institute of Food Science, Food Technology and Nutrition (ICTAN), Spanish National Research Council (CSIC), Jose Antonio Novais 10, 28040 Madrid, Spain; 2Research Institute Juan de Dios Guevara, Faculty of Pharmacy and Biochemistry, Major National University of San Marcos, Lima 15021, Peru; 3Zootecnia an Animal Production Laboratory, Faculty of Veterinary Medicine, Major National University of San Marcos, Lima 15021, Peru; 4Research Institute IVITA-Pucallpa, Faculty of Veterinary Medicine, Major National University of San Marcos, Lima 15021, Peru; 5Functional Products Laboratory, Faculty of Engineering in Food Industries, National University of Central Peru, Huancayo 12006, Peru; 6Pharmacology and Toxicology Laboratory, Faculty of Veterinary Medicine, Major National University of San Marcos, Lima 15021, Peru

**Keywords:** *Brunfelsia grandiflora*, medicinal plant, polyphenols, hydroxycoumarins, antioxidant activity, LC-MS-QToF

## Abstract

*Brunfelsia grandiflora* is an ancient plant widely used for its promising medicinal properties, although little explored scientifically. Despite being a rich source of phenolic compounds responsible in part for the proven anti-inflammatory activity, its characterization has not been carried out to date. The present work deals with the exhaustive identification and quantification of its phenolic fraction, along with its antioxidant activity. Decoction resulting from the bark as fine powder was filtered and lyophilized, and polyphenols were extracted from the resulting product by aqueous-organic solvents. Seventy-nine polyphenols were identified using LC-MS^n^. Hydroxycinnamates was the most abundant group of compounds (up to 66.8%), followed by hydroxycoumarins (15.5%), lignans (6.1%), flavonols (5.7%), phenolic simples (3.1), gallates (2.3%), flavanols (0.3%), and flavanones (0.2%). About 64% of the characterized phenols were in their glycosylated forms. The quantification of these phytochemicals by LC-QToF showed that this medicinal plant contained 2014.71 mg of phenolic compounds in 100 g dry matter, which evidences a great antioxidant potency determined by ABTS and DPPH assays. Therefore, *Brunfelsia grandiflora* represents an important source of polyphenols which supports its therapeutic properties scientifically proven.

## 1. Introduction

Healing with medicinal plants is as old as humanity itself. Awareness of medicinal plant usage is a result of the many years of struggles against illnesses due to which man learned to pursue drugs in barks, seeds, fruit bodies, and other parts of the plants. The need to integrate the knowledge of traditional medicine with scientific medicine, based on experience and observation, makes it necessary to validate therapeutic action and establish the correct uses of plant resources. This is the case of *Brunfelsia glandiflora*, a traditional native remedy employed against rheumatism, arthritis, fevers, and snake bites in the upper Amazon region [1]. *Brunfelsia glandiflora* is a plant belonging to the *Solanaceae* family and the *Brunfelsia* genus, traditionally known as Chiric sanango, chiricaspi chacrudo; hu-ha-hai, sanango, mucapari, and chirihuayusa [2].

*Brunfelsia glandiflora* is a glabrous shrub up to five meters high, with tough bark, alternate leaves, apically leafy or scattered on flowering branches, 15–20 cm long, 5–8 cm wide. It has cymose inflorescence, pedicellate flowers 3.5–4 cm long, which are purple and white with tubular, campanulate corolla with five large lobes, and short calyx 1.5–2 cm long. Anthers are free from the stigma, small, obtuse, appendicular at the base, and superior bicarpelar ovary (Figure 1). Fruit in berry is ovate-rounded. It grows in the Andean mountainous area between Venezuela and Bolivia. It is distributed at the height of 200 m in Peru, above sea level, in the low and high Amazon areas (Regions of Loreto, Ucayali, Madre de Dios, and Cuzco) [2,3]. *Brunfelsia grandiflora* species known as “Chiric Sanango” is mainly sold in the medicinal plant markets, especially in the Amazon regions and in the capital Lima, from wild populations or home gardens, similar to other medicinal species. Our ancestors commonly used woody vascular plants, mainly their bark, and today this part of the plant is renowned as a source of antioxidants with potential health-promoting properties.

There are few scientific publications on the pharmacological action of *B. grandiflora*, but the one described in the traditional medicine of the Peruvian Amazon refers to the aqueous maceration of the root of *Brunfelsia grandiflora*, which is used as a drink against arthritis, syphilis, bone pain, ovarian pain, fatigue and as an antipyretic. The infusion of the leaves against arthritis and rheumatism is another form of common use. Some reports mention that the bark decoction is applied to burns, to areas of the body affected by leishmaniasis, and as a healing agent, although its narcotic effects have also been reported [1,2,4,5].

The few pharmacological effects of *B. grandiflora* described above could be due to the presence of secondary metabolites such as polyphenolic acid compounds. One of these compounds could be scopoletin [6], with known anti-inflammatory activity, which would justify the effect of *B. grandiflora* against rheumatism, arthritis, body pain, headache, and joint and muscle pain. On the other hand, the hallucinogenic and narcotic properties associated with *B. grandiflora* would be mediated by brunfelsamidine, cuscohygrin, scopolamine, scopoletin, and esculetin, this one last used in oncology as an antiproliferative. Furthermore, the effects of brunfelsamidine and cuscohygrine in the fields of anesthesiology have been demonstrated [7,8,9].

The main objective of this work was to identify for the first time the phenolic composition of this medicinal plant to know the chemical structures of these phytochemicals that are behind the renowned biological properties of *Brunfelsia grandiflora*. Additionally, polyphenol content and antioxidant capacity will be determined to evaluate the magnitude of this phytochemical fraction in *Brunfelsia grandiflora*.

## 2. Results and Discussion

Ever since ancient times, people have looked for drugs in nature to face different diseases. *Brunfelsia grandiflora* is an excellent example of folk medicine used for ages with successful results against rheumatism, arthritis, cold, tiredness, pain of ovaries, sexual potency, pain in bones, laziness, and cancer of uterus [2], although limited scientific studies confirm these effects [10,11]. Even in our time, when there is increasing awareness of the importance of diet quality to prevent chronic disease, and although the main sources of phenolic compounds are fruits and vegetables, more and more studies refer to woody vascular plants, especially bark [12], directing the interest to the traditional herbal as a source of antioxidants with potential health-promoting properties. This situation points out the importance of considering these medicinal plants as an adjuvant to deal with prevalent diseases and hence the adequacy of properly characterizing their phytochemical composition. Phenolic compounds are ubiquitously distributed phytochemicals found in most plant sources with recognized health benefits [13], and as far as we know, the phenolic fraction in *Brunfelsia grandiflora* has never been characterized. In the present work, the identification and quantification of the phenolic fraction were assessed in a lyophilized extract obtained from the *Brunfelsia grandiflora* bark. Additionally, total phenolic content by Folin–Ciocalteu and the antioxidant capacity was carried out.

### 2.1. Total Phenolic Content and Antioxidant Capacity

The total phenolic content determined by the Folin–Ciocalteu assay and its antioxidant potency developed by FRAP and DPPH assays are summarized in Table 1. IC_50_ values determined by both FRAP and DPPH assays are included. The evaluated extract had about 3% of the phenolic content of the dry matter. Additionally, the antioxidant ability of the *Brunfelsia grandiflora* bark was tested by two methods (DPPH and ABTS) that measure the ability of antioxidants contained in this medicinal plant to scavenge the DPPH and ABTS, respectively, and based on an electron transfer and the reduction of a colored oxidant. The IC_50_ (half-maximal inhibitory concentration) was calculated as the concentration of sample necessary to decrease by 50% the initial absorbance of DPPH and ABTS. Both methods showed a very high radical scavenging, 2.55 and 4.55 μg/mL for DPPH and ABTS, respectively. These values agree with the high polyphenol amount determined with the spectrophotometric Folin–Ciocalteu method. Recently, the antioxidant capacity of an herbal remedy (HR) was compared with that of a crude hydroalcoholic extract (CHE) obtained from *Brunfelsia uniflora* (Pohl) D. Don roots [14]. IC_50_ values determined by the ABTS assay showed significantly higher values (1678.00 ± 11.26 μg/mL and 3441.00 ± 36.05 μg/mL for HR and CHE, respectively) than that determined for *Brunfelsia grandiflora* bark (4.55 μg/mL). Likewise occurred with the IC_50_ determined by DPPH for HR and CHE, where the values of 37,698.00 ± 3437.00 μg/mL and 68,452.00 ± 5155.00 μg/mL of HR and CHE, respectively, were much higher than that obtained for *Brunfelsia grandiflora* (2.55 μg/mL), which suggested a substantially higher antioxidant activity of our medicinal plant than that evaluated in this article. Borneo et al. [15] characterized the antioxidant capacity by DPPH of 15 *Asteraceae* plant species from Cordoba (Argentina) in relation to their phenol content determined by the Folin–Ciocalteu assay. Phenolic content ranged from 11.3 to 54.4 mg/g, and their IC_50_ values from 198 to 2009 μg/mL, which were higher than that determined for ascorbic acid, BHT, and quercetin (11.5, 15.3, and 14.8 μg/mL, respectively). *Brunfelsia grandiflora* showed higher antioxidant potency than the Argentinian plants and, more importantly, well-known antioxidants such as ascorbic acid, BHT, and quercetin. A recent study developed by Rebolledo et al. [16] with the Peruvian peppertree *Schinus* areira L. from Chile observed that the methanolic extracts were highly rich in both polyphenols (>195 mg/g dw~19.5%) and antioxidant activity (IC_50_ > 476 mg/mL; >273 mg ascorbic acid/g dw (DPPH); >301 mg ascorbic acid/g dw (FRAP)) and were in line with that described in the present manuscript. Therefore, the high antioxidant potency of *Brunfelsia grandiflora* bark highlights the potential of this plant for pharmacological use.

### 2.2. LC-QToF Identification of the Phenolic Fraction of Brunfelsia grandiflora

Seventy-six phenolic compounds were identified in *Brunfelsia grandiflora* based on their relative retention time, mass spectra and commercial standards. Table 2 shows the retention time (RT), molecular formula, accurate mass of the molecular ion [M − H]^−^ after negative ionization, and MS^2^ fragments of the main compounds identified in *Brunfelsia grandiflora* by LC-QToF.

The presence of scopoletin has been mentioned in the few studies carried out with *Brunfelsia grandiflora* [6], which belong to hydroxycoumarin group. This compound was identified thanks to its MS spectra ([M − H]^−^ at *m*/*z* 191.0350 and fragment ions at *m*/*z* 148, 120, and 104). Also belonging to coumarins, it was identified esculetin and its glycosylated derivative (esculin) at 5.7 and 3.7 min, respectively. The first one showed a quasimolecular ion at *m*/*z* 177.0193 and fragment ions at *m*/*z* 149, 133, and 105 compatible with its chemical structure, and the glycosylated derivative ([M − H]^−^ at *m*/*z* 339.0722) showed as the main fragment ion that corresponds to its free precursor, esculetin (*m*/*z* 177) (Table 2).

Some of the identified compounds showed a chemical structure belonging to gallates, such as methyl- and ethyl-gallate, as well as galloyl glucose. Three isomers of methyl-gallates at 3.6, 6.6, and 9.4 min showed a [M − H]^−^ at *m*/*z* 183.0299 and fragmented ions at *m*/*z* 168 and 124. Likewise, four isomers of ethyl-gallate at 6.3, 7.2, 11.4, and 13.0 min were identified based on their compatible MS spectra ([M − H]^−^ at *m*/*z* 197.0455 and fragment ions at *m*/*z* 169 and 124). Only one chromatographic peak showed MS spectra suited with galloyl-glucose at 5.0 min ([M − H]^−^ at *m*/*z* 331.0671 and fragment ion at *m*/*z* 169). Finally, gallic acid was unambiguously identified at 2.0 min, thanks to its commercial standard and the MS spectra (Table 2).

An important group of phenolic compounds identified in *Brunfelsia grandiflora* belonged to hydroxycinnamic acids. Three isomers of caffeoylquinic acids were identified at 4.7, 5.0, and 5.1 min due to their MS spectra ([M − H]^−^ at *m*/*z* 353.0878 and fragment ion at *m*/*z* 191, characteristic of quinic acid). The earliest chromatographic peak was assigned to 5-caffeoylquinic acid (chlorogenic acid) thanks to the commercial standard. More hydroxycinnamic acids esterified with quinic acid were identified in this plant. Coumaroyl- and sinapoylquinic acids, along with two isomers of feruloylquinic acid were identified at 12.8, 13.5, 13.4, and 13.7 min, respectively, due to their respective quasimolecular ions at *m*/*z* 337.0929, 397.1140 and 367.1035, respectively. MS/MS allowed confirming their identity after determining their respective precursor at *m*/*z* 163, 223, and 193, respectively. Related to the described hydroxycinnamates, it was identified both their free hydroxycinnamic acids and glycosylated forms. Thus, caffeic acid ([M − H]^−^ at *m*/*z* 179.0350 and fragment ion at *m*/*z* 135) as well as caffeic acid-*O*-glucoside ([M − H]^−^ at *m*/*z* 341.0878 and fragment ions at *m*/*z* 179 and 161) appeared at 5.9 and 11.3 min, respectively. Ferulic acid at 9.0 min ([M − H]^−^ at *m*/*z* 193.0506 and fragment ions at *m*/*z* 134 and 149) and three isomers of ferulic acid-*O*-glucoside at 13.0, 13.4, and 13.8 min ([M − H]^−^ at *m*/*z* 355.1035 and fragment ions at *m*/*z* 193, 149 and 134) were identified in *Brunfelsia grandiflora*. Coumaric acid at 8.0 min ([M − H]^−^ at *m*/*z* 163.0401 and fragment ion at *m*/*z* 119), three isomers of coumaric acid-*O*-glucoside at 12.5, 15.6, and 17.4 min ([M − H]^−^ at *m*/*z* 325.0929 and fragment ions at *m*/*z* 163, 119), sinapic acid at 9.4 min ([M − H]^−^ at *m*/*z* 223.0612 and fragment ions at *m*/*z* 193, 149 and 165) and two isomers of sinapic acid-*O*-glucoside at 13.1 and 15.2 min ([M − H]^−^ at *m*/*z* 385.1140 and fragment ions at *m*/*z* 223 and 149) were characterized. Regarding ferulic acid, dehydrodiferulic acid was also identified in this extract based on its quasimolecular ion at *m*/*z* 385.0929 and fragment ions at *m*/*z* 193, 177, and 149, and its chromatographic peak eluted at 10.2 min (Table 2).

A minor group of the identified compounds corresponded to flavanols, such as gallocatechin and methyl-gallocatechin. Both commercial standards and MS spectra allowed their unequivocal identification at 6.1 and 14.2 min, respectively (Table 2).

Seven flavanones were also identified in the lyophilized extract from *Brunfelsia grandiflora* bark. Eriodictyol and two glycosylated derivatives were characterized at 10.2, 12.3, and 16.2 min, respectively. Commercial eriodictyol facilized its identification along with its MS spectra, while eriodyctiol-*O*-glucoside showed well-suited MS spectra ([M − H]^−^ at *m*/*z* 449.1089 and fragment ions at *m*/*z* 287 and 255 corresponding to eriodictyol). Likewise, naringenin and two glycosylated derivatives at 15.9, 12.1, and 18.0 min, respectively, were characterized based on their compatible MS spectra. Hesperetin was also present in *Brunfelsia grandiflora* bark, although no glycosylated derivative was identified.

Belonging to the flavonols group, two compounds with the same molecular formula (C_27_H_30_O_15_) were identified as kaempherol-*O*-rutinoside and kaempherol-*O*-galactoside-rhamnoside based on their different fragmentation pattern (Table 2). In addition, they eluted at 4.9 and 12.0 min, respectively, in agreement with their polar nature. Likewise happened with isorhamnetin-*O*-rutinoside and isorhamnetin-*O*-glucoside-*O*-rhamnoside, although they showed the same quasimolecular ion at *m*/*z* 623.1618 and fragment ion at *m*/*z* 315 corresponding to isorhamnetin, the greater polarity of isorhamnetin-*O*-rutinoside allowed it to be associated with the chromatographic peak that eluted at 10.6 min, and isorhamnetin-*O*-glucoside-*O*-rhamnoside with that which eluted at 12.2 min.

The phenolic fraction of *Brunfelsia grandiflora* was also constituted by lignans, such as pinoresinol and matairesinol. These phenolic compounds presented the same molecular formula and, therefore, equal quasimolecular ion at *m*/*z* 357.1344, and, unfortunately, MS/MS analysis showed no fragment ion. Nevertheless, their different polar nature allowed us to know that pinoresinol eluted at 8.8 min while matairesinol at 9.5 min. Two isomers of hydroxysecoisolariciresinol were identified at 9.6 min and 9.9 min based on their MS analysis ([M − H]^−^ at *m*/*z* 377.1606 and fragment ion at *m*/*z* 329). Secoisolariciresinol, along with two isomers, were characterized thanks to their MS analysis, which eluted between 9.8 and 15.2 min (Table 2). Sesamol with a quasimolecular ion at *m*/*z* 137.0244 was associated with the chromatographic peak eluting at 11.3 min. The isomers sesamin and episesamin eluted at 13.4 and 19.4 min, respectively, and showed a [M − H]^−^ at *m*/*z* 353.1031 and fragment ion at *m*/*z* 96, compatible with their chemical structure. The chromatographic peak at 12.7 min showed MS spectra ([M − H]^−^ at *m*/*z* 359.15 and fragment ion at *m*/*z* 313), and the molecular formula (C_20_H_24_O_6_) compatible with cyclolariciresinol and isolariciresinol, which did not allow us to determine the identity of the lignin. Likewise, the chromatographic peak at 12.9 min showed MS spectra compatible with hydroxymatairesinol and nortrachelogenin.

Finally, simple phenolic acids were also characterized in this medicinal plant, most of them supported by commercial standards such as protocatechuic acid, 3- and 4-hydroxybenzoic acid, vanillic acid, homovanillic acid, 3,4-dihydroxyphenylpropionic acid, 3- and 4-hydroxyphenylpropionic acids and 3-methoxy-4-hydroxyphenylpropionic acid. Additionally, three isomers of dihydroxybenzoic acid, two isomers of methoxy-hydroxybenzoic acid, two isomers hydroxyphenylacetic acid, and the other isomer of methoxy-hydroxyphenylpropionic acid were characterized based on their respective MS spectra (Table 2). This group was completed with the characterization of glycosylated derivatives of dihydroxybenzoic acid and methoxy-hydroxybenzoic acid, thanks to the quasimolecular ion at *m*/*z* 315.0722 and 329.0878, respectively, and the presence of their precursor, benzoic acid, among the fragment ions.

### 2.3. LC-QToF Quantification of the Phenolic Content of Brunfelsia grandiflora

The quantification of these phytochemicals by LC-QToF showed that *Brunfelsia grandiflora* contained 2014.71 mg of phenolic compounds in 100 g dry matter. This amount is lower than that determined by the Folin–Ciocalteu assay (3017 mg/100 g dry matter) but coherent because it is well-known that the Folin–Ciocalteu assay could over-estimate the real polyphenol content. It is impossible to compare with data reported in the literature because this is the first time that the phenolic fraction of *Brunfelsia grandiflora* is characterized.

The most abundant group of polyphenols present in *Brunfelsia grandiflora* was composed of hydroxycinnamic acids, which amounted to 66,8% of the total phenols quantified. These compounds were preferentially present as hydroxycinnamates, either esterified with glucose to form glycosidic derivatives of caffeic, ferulic, coumaric, and sinapic acids (86.2% of the total hydroxycinnamic acids) or with quinic acid to form hydroxycinnamoyl derivatives such as caffeoyl-, feruloyl, coumaroyl- and sinapoylquinic acids (12.9% of the total hydroxycinnamic group). The free precursors, along with dehydrodiferulic acid, barely represented 0.9% of the total hydroxycinnamic acids (Table 3).

The following compound group characterized in *Brunfelsia grandiflora* by order of abundance was that corresponding to hydroxycoumarins (15.5% of the total phenolic content), led by scopoletin (91.6% of the total of this group) and followed by esculetin and esculin (Table 3).

The following more abundant compound group was lignans (6.1% of the total phenolic content), led by sesamol and sesamin/episesamin, with 44.8% and 17.5%, respectively, of the total of this group. Secoisolariciresinol and hydroxysecoisolariciresinol were also predominant, amounting to 14.9% of the total lignans (Table 3).

Flavonols were the next most abundant compound group (5.7% of the total phenolic content), headed by isorhamnetin-*O*-glucoside-*O*-rhamnoside and followed by isorhamnetin-*O*-rutinoside, kampherol-*O*-galactoside-*O*-rhamnoside, and kampherol-*O*-rutinoside (82.7, 7.1, 6.7 and 3.5%, respectively, of the total flavonols) (Table 3).

The following group was that corresponding to phenolic acids, widely distributed in vegetable sources, reaching 3.1% of the total phenolic content. Although these compounds were preferentially in their free forms, glycosylated forms of dihydroxybenzoic and methoxy-hydroxybenzoic acids represented 13.8% of phenolic acids characterized. The top five most abundant compounds were two isomers of hydroxyphenylacetic acid, protocatechuic acid, methoxy-hydroxybenzoic acid, and 3-hydroxybenzoic acid (Table 3).

Gallates group represented 2.3% of the total phenolic fraction. This group was headed by ethyl-gallate followed by methyl-gallate, accounting for 85.1% and 8.9% of the total gallates, respectively. The remaining 6.0% was composed of galloyl glucose and free gallic acid (Table 3).

Gallocatechin and methyl-gallocatechin belonging to flavanols were also present in *Brunfelsia grandiflora* (0.29% of the total phenolic content), headed by gallocatechin (95.0% of this group) (Table 3).

Preferentially glycosidic forms of flavanones eriodictyol, naringenin, along with their free precursors, and hesperetin represented 0.21% of the total phenolic content of *Brunfelsia grandiflora*. Approximately half was comprised of eriodictyol and derivatives, and the other half of naringenin and derivatives, while hesperetin barely reached 2% of the total of this specific fraction.

Regarding the correlation between chemical composition and antioxidant capacity, it is worth noting that the massive content of hydroxycinnamic acids/hydroxycinnamates in *B. grandiflora* is enough to grant a remarkable antioxidant power, as we have previously reported in vitro and cell culture [17,18,19,20,21,22,23]. It is well known the correlation of the antioxidant activity of polyphenols with the number and position of -OH groups or the presence of a double bond in the position 2–3 of C ring in flavonoids. Likewise, the antioxidant activity of polyphenolic acids depends on the number of -OH groups in their molecule. Thus, gallic acid, caffeic acid, catechin, and eriodictyol and their derivatives will strongly contribute to the antioxidant potency of this plant. Likewise, dihydroxybenzoic, dihydroxyphenylacetic, and dihydroxyphenylpropionic acids are also key antioxidants present in this medicinal plant. Further, this antioxidant capability has translated into beneficial biological effects in experimental animal models [24]. We are currently investigating the effect of B. grandiflora extracts on cultured endothelial EA.hy926 and neuronal SH-SY5Y cells submitted to oxidative stress to confirm the chemo-protective potential of extracts from the bark of this plant to explain and sustain its traditional medicinal utilization. Thus, this research should be considered as a starting point for a series of studies devoted to proving the cellular and molecular basis that supports the medicinal use of this plant.

## 3. Materials and Methods

### 3.1. Chemical Reagents

Bark of *Brunfelsia grandiflora* was collected from the native community of Canaán de Cachiyacú, Contamana district, Ucayali province, Loreto region (Peru). All solvents and reagents were of analytical grade unless otherwise stated. Gallic acid, 3,4-dihydroxyphenylpropionic acid, 4-hydroxy-3-methoxyphenylpropionic acid, 3- and 4-hydroxybenzoic acids, 4-hydroxyphenylacetic acid, protocatechuic acid, 4-hydroxyphenylpropionic acid, 4-hydroxy-3-methoxyacetic acid (homovanillic acid), 5-caffeoylquinic acid, caffeic acid, ferulic acid, *p*-coumaric acid, sinapic acid, gallocatechin, hesperetin, eriodictyol, naringenin, isorhamentin, 2,2-diphenyl-1-picrylhydrazyl (DPPH), and trolox (6-hydroxy-2,5,7,8-tetramethylchromo-2-carboxylic acid) (97%) were acquired from Sigma-Aldrich (Madrid, Spain). Formic acid and acetonitrile (HPLC grade) were acquired from Panreac (Madrid, Spain).

### 3.2. Sample Preparation

The bark was wholly collected and washed. It was dried in the open air to constant weight and reduced to a fine powder. The decoction was developed by placing distilled water and powdered plant material (10:1) in a beaker, heating to boiling, and holding for twenty minutes. The plant material exhausted by the extraction was separated by filtration, and the aqueous extract was concentrated and lyophilized for preservation.

To determine total phenolic content by Folin–Ciacalteu and antioxidant activity by DPPH and ABTS assays, a methanolic solution from the lyophilized extract obtained from the *Brunfelsia grandiflora* bark was prepared at 0.02 g/mL and diluting later with water up to 0.8 mg/mL.

To determine the antioxidant activity by DPPH and ABTS assays, the sample was dissolved in ethanol 96% (*v*/*v*) to obtain concentrations from 4 to 16 μg/mL and from 2 to 8 μg/mL, respectively.

The procedure of Perez-Jimenez et al. [25] was followed with minor modifications to isolate polyphenols from the bark of *Brunfelsia grandiflora* in order to characterize them by LC-ESI-QTOF. Briefly, 0.25 g by quadruplicate of the lyophilized extract was extracted in aqueous methanol (50:50, *v*/*v*, with HCl 2 N, 1 h) by constant shaking and centrifuged at 3000× *g*. Supernatants were separated, and the pellets were washed with acetone/water (70:30, *v*/*v*) by constant shaking and centrifuged at 3000× *g*. Supernatants from each extraction step were combined at 50 mL. An aliquot of 1 mL was concentrated under reduced pressure using a vacuum concentrator system (Speed-Vac, Thermo Fisher Scientific Inc., Waltham, MA, USA) and then resuspended in 0.5 mL of 1% formic acid in deionized water (*v*/*v*), filtered through a cellulose-acetate membrane filter of 0.45 μM pore-size, dispensed in chromatographic vials and stored at −80 °C until analysis.

### 3.3. Polyphenolic Content by Folin–Ciocalteu

The total phenolic content of *Brunfelsia grandiflora* bark was quantified spectrophotometrically at 765 nm using the Folin–Ciocalteu reagent following ISO 14502-1 procedure [26]. Then, 100 μL of the methanolic extract of the *Brunfelsia grandiflora was* prepared as described above in Section 3.2. The section was mixed with 500 μL of Folin-Ciocalteu diluted with water (1:9, *v*/*v*) and let stand for 5 min. Then, 400 μL of Na_2_CO_3_ 7.5% *w/v* was added and shaken vigorously. After 1 h incubation at room temperature (25 °C), the absorbance was measured in a spectrophotometer (Thermo Scientific, Waltham, MA, USA) at 765 nm. Gallic acid was used as standard, and results were expressed as mg gallic acid equivalents (GAE) per 100 g of dry matter.

### 3.4. Determination of Antioxidant Capacity

The antioxidant capacity of *Brunfelsia grandiflora* extracts prepared as described in Section 3.2. was determined by two different methods.

DPPH• radical scavenging assay: the stable free radical 2,2-diphenyl-1-picrylhydrazyl (DPPH) was used to evaluate the radical scavenging activity of the samples, following the method reported by Brand-Williams et al. [27] and Thaipong et al. [28]. The stable free radical DPPH• is purple and is discolored to yellow in the presence of a free radical-capturing substance whose measurement at 517 nm (spectrophotometer GENESYS^TM^ 10S UV-VIS) is related to the antioxidant capacity of the substance. Trolox was taken as a reference, and the results were expressed as mg Trolox Equivalent Antioxidant Capacity (TEAC) per gram of dry matter. IC_50_ was also determined and expressed as μg/mL.

ABTS assay: the free radical cation ABTS+, which was prepared by reaction of ABTS with 2.45 mM potassium persulfate during 12–16 h at room temperature in the dark, was used to evaluate the free radical scavenging capacity of the samples. This radical decreases absorbance at 734 nm in the presence of an antioxidant [28,29]. The absorbance was monitored for 30 min at 37 °C in a spectrophotometer GENESYSTM 10S UV-VIS. Results were expressed as mg TEAC per gram of dry matter. IC50 was also determined and expressed as μg/mL.

### 3.5. Phenolic Characterization of Brunfelsia grandiflora by LC-ESI-QTOF Analysis

Phenolic compounds from *Brunfelsia grandiflora* were characterized by HPLC-ESI-QToF [30] in an Agilent 1200 series LC system coupled to an Agilent 6530A Accurate-Mass Quadrupole Time-of-Flight (Q-ToF) with ESI-Jet Stream Technology (Agilent Technologies). Compounds were separated on a reverse-phase InfinityLab Poroshell 120 EC-C18 (15 cm × 3 mm, 2.7 μm) column (Agilent Technologies) preceded by a guard column (3 × 5 mm × 2.7 μm) at 40 °C. Then, 10 μL of the sample was injected and separated by using a mobile phase consisting of Milli-Q water (phase A) and acetonitrile (phase B), both containing 0.1% formic acid, at a flow rate of 0.5 mL/min. The mobile phase was initially programmed with 90% of solvent A and 10% of B. The elution program increased to 30% of solvent B in 10 min, 40% solvent B in 5 min, and 50% of solvent B in 5 min. Then, the initial conditions (10% solvent B) were recovered in 5 min and maintained for 5 min. The Q-ToF acquisition conditions were as follows: drying gas flow (nitrogen, purity > 99.9%) and temperature were 10 L/min and 325 °C, respectively; sheath gas flow and temperature were 6 L/min and 250 °C, respectively; nebulizer pressure was 25 psi; cap voltage was 3500 V, and nozzle voltage was 500 V. Mass range selected was from 100 up to 970 m/z in negative mode and fragmentor voltage of 150 V. Data were processed in a Mass Hunter Workstation Software. External calibration curves were prepared with the following standards (gallic acid, 3,4-dihydroxyphenylpropionic acid, 4-hydroxy-3-methoxyphenylpropionic acid, 3- and 4-hydroxybenzoic acids, 4-hydroxyphenylacetic acid, protocatechuic acid, 4-hydroxyphenylpropionic acid, 4-hydroxy-3-methoxyacetic acid (homovanillic acid), 5-caffeoylquinic acid, caffeic acid, ferulic acid, *p*-coumaric acid, sinapic acid, gallocatechin, hesperetin, eriodictyol, naringenin, isorhamentin) at five different concentration levels from 0.001 to 20 μM. Limit of detection and quantification ranged from 0.002 to 0.006 μM and from 0.004 to 0.007 mM, respectively. The inter- and intra-day precision of the assay (as the coefficient of variation, ranging from 3.8 to 7.9%) was considered acceptable and allowed the quantification of phenolic compounds (quantified as equivalents of the respective parent molecules when they were available or the most chemically related).

## 4. Conclusions

This work demonstrates that *Brunfelsia grandiflora* represents an important source of polyphenols (2% dry matter) and, therefore, antioxidant activity. Up to seventy-nine polyphenols were characterized for the first time, which belonged to hydroxycinnamates, hydroxycoumarins, lignans, flavonols, gallates, flavanols, flavanones, and phenolic acids. Knowing the phenolic composition of *Brunfelsia grandiflora* and the antioxidant capacity of its bark extracts will be useful for the design of future cell culture studies and experimental designs in animals to understand why our ancestors used this medicinal plant.

## Figures and Tables

**Figure 1 molecules-27-06510-f001:**
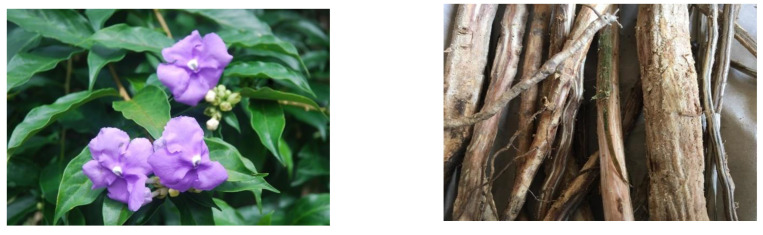
*Brunfelsia grandiflora* from Perú. On the right is the plant in the flowering period (photo courtesy of Jana Horackova), and on the left is the bark used to obtain the extract used in the present study.

**Table 1 molecules-27-06510-t001:** Total polyphenolic content and antioxidant capacity of lyophilized extract obtained from *Brunfelsia grandiflora*.

*Brunfelsia grandiflora*	
Total Phenolic content by Folin (g/100 g d.m.)	3.02 ± 0.33
DPPH (mg Trolox/g d.m.)	11.86 ± 1.45
ABTS (mg Trolox/g d.m.)	80.38 ± 4.22
IC_50_ by DPPH (μg/mL)	2.55 ± 0.12
IC_50_ by ABTS (μg/mL)	4.55 ± 0.10

Values are the average ± standard deviation of three different samples and expressed on a dry matter (d.m.) basis.

**Table 2 molecules-27-06510-t002:** LC-QToF identification of phenolic compounds of *Brunfelsia grandiflora*.

Identified Compound	RT (min)	Molecular Formula	Molecular Weight	[M − H]^−^	Fragment MS^2^
**HYDROXYCOUMARINS**					
Esculin	3.7	C_15_H_16_O_9_	340.0794	339.0722	177; 133
Esculetin	5.7	C_9_H_6_O_4_	178.0266	177.0193	133; 105; 149
Scopoletin	8.9	C_10_H_8_O_4_	192.0423	191.0350	104; 120; 148
**GALLATES**					
Gallic acid	2.0	C_7_H_6_O_5_	170.0215	169.0142	125
Methyl-gallate	3.6	C_8_H_8_O_5_	184.0372	183.0299	168; 124
Galloyl-glucose	5.0	C_13_H_16_O_10_	332.0743	331.0671	169
Methyl-gallate	6.6	C_8_H_8_O_5_	184.0372	183.0299	124
Ethyl-gallate	6.3	C_9_H_10_O_5_	198.0528	197.0455	169; 124
Ethyl-gallate	7.2	C_9_H_10_O_5_	198.0528	197.0455	169; 124
Methyl-gallate	9.4	C_8_H_8_O_5_	184.0372	183.0299	168
Ethyl-gallate	11.4	C_9_H_10_O_5_	198.0528	197.0455	169; 124
Ethyl-gallate	13.0	C_9_H_10_O_5_	198.0528	197.0455	169; 124
**HYDROXYCINNAMIC ACIDS AND HYDROXYCINNAMATES**					
5-Caffeoylquinic acid	4.7	C_16_H_18_O_9_	354.0951	353.0878	191; 93; 173
Caffeoylquinic acid	5.0	C_16_H_18_O_9_	354.0951	353.0878	191
Caffeoylquinic acid	5.1	C_16_H_18_O_9_	354.0951	353.0878	191
Caffeic acid	5.9	C_9_H_8_O_4_	180.0423	179.0350	135
Coumaric acid	8.0	C_9_H_8_O_3_	164.0473	163.0401	119
Ferulic acid	9.0	C_10_H_10_O_4_	194.0579	193.0506	134; 149
Sinapic acid	9.4	C_11_H_12_O_5_	224.0685	223.0612	193, 149, 165
Dehydrodiferulic acid	10.2	C_20_H_18_O_8_	386.1002	385.0929	193; 177; 149
Caffeic acid-*O*-glucoside	11.3	C_15_H_18_O_9_	342.0951	341.0878	179; 161
Coumaric acid-*O*-glucoside	12.5	C_15_H_18_O_8_	326.1002	325.0929	163; 119
Coumaroylquinic acid	12.8	C_16_H_18_O_8_	338.1002	337.0929	191; 163
Ferulic acid-*O*-glucoside	13.0	C_16_H_20_O_9_	356.1107	355.1035	193; 149; 134
Sinapic acid-*O*-glucoside	13.1	C_17_H_22_O_10_	386.1213	385.1140	223
Ferulic acid-*O*-glucoside	13.4	C_16_H_20_O_9_	356.1107	355.1035	193; 134
Feruloylquinic acid	13.4	C_17_H_20_O_9_	368.1107	367.1035	191; 193
Sinapoylquinic acid	13.5	C_18_H_22_O_10_	398.1213	397.1140	223
Feruloylquinic acid	13.7	C_17_H_20_O_9_	368.1107	367.1035	193; 191
Ferulic acid-*O*-glucoside	13.8	C_16_H_20_O_9_	356.1107	355.1035	193; 134
Sinapic acid-*O*-glucoside	15.2	C_17_H_22_O_10_	386.1213	385.1140	223; 149
Coumaric acid-*O*-glucoside	15.6	C_15_H_18_O_8_	326.1002	325.0929	163
Coumaric acid-*O*-glucoside	17.4	C_15_H_18_O_8_	326.1002	325.0929	119
**FLAVANOLS**					
Gallocatechin	6.1	C_15_H_14_O_7_	306.0740	305.0667	125; 137
Methyl-epigallocatechin	14.2	C_16_H_16_O_7_	320.0896	319.0823	275; 137
**FLAVANONES**					
Eriodictyol	10.2	C_15_H_12_O_6_	288.0634	287.0561	285; 283; 287; 255
Naringenin-*O*-glucoside	12.1	C_21_H_22_O_10_	434.1213	433.1140	271; 151
Eriodictyol-*O*-glucoside	12.3	C_21_H_22_O_11_	450.1162	449.1089	287; 255
Naringenin	15.9	C_15_H_12_O_5_	272.0685	271.0612	151; 177
Eriodictyol-*O*-glucoside	16.2	C_21_H_22_O_11_	450.1162	449.1089	287
Hesperetin	16.9	C_16_H_14_O_6_	302.0790	301.0718	286; 242
Naringenin-*O*-glucoside	18.0	C_21_H_22_O_10_	434.1213	433.1140	271
**FLAVONOLS**					
Kaempherol-*O*-rutinoside	4.9	C_27_H_30_O_15_	594.1585	593.1512	284, 285, 255
Isorhamnetin-*O*-rutinoside	10.6	C_28_H_32_O_16_	624.1690	623.1618	315
Kaempherol-*O*-galactoside-*O*-rhamnoside	12.0	C_27_H_30_O_15_	594.1585	593.1512	285; 257; 284
Isorhamnetin-*O*-glucoside-*O*-rhamnoside	12.2	C_28_H_32_O_16_	624.1690	623.1618	315
**LIGNANS**					
Pinoresinol	8.8	C_20_H_22_O_6_	358.1416	357.1344	N.D.
Matairesinol	9.5	C_20_H_22_O_6_	358.1416	357.1344	N.D.
Hydroxysecoisolariciresinol isomer	9.6	C_20_H_26_O_7_	378.1679	377.1606	329
Secoisolariciresinol isomer	9.8	C_20_H_26_O_6_	362.1729	361.1657	165
Hydroxysecoisolariciresinol isomer	9.9	C_20_H_26_O_7_	378.1679	377.1606	329
Sesamol	11.3	C_7_H_6_O_3_	138.0317	137.0244	N.D.
Secoisolariciresinol	11.8	C_20_H_26_O_6_	362.1729	361.1657	346; 165
Cyclolariciresinol or Isolariciresinol	12.7	C_20_H_24_O_6_	360.1573	359.15	313
Hydroxymatairesinol/Nortrachelogenin	12.9	C_20_H_22_O_7_	374.1366	373.1293	355
Sesamin	13.4	C_20_H_18_O_6_	354.1103	353.1031	96
Secoisolariciresinol Isomer	15.2	C_20_H_26_O_6_	362.1729	361.1657	165
Episesamin	19.4	C_20_H_18_O_6_	354.1103	353.1031	96
**OTHER PHENOLIC ACIDS**					
Methoxy-hydroxybenzoic acid glucoside	2.3	C_14_H_18_O_9_	330.0951	329.0878	167; 108
Dihydroxybenzoic acid glucose	2.4	C_13_H_16_O_9_	316.0794	315.0722	153; 109
Dihydroxybenzoic acid glucose	2.6	C_13_H_16_O_9_	316.0794	315.0722	153; 109
3,4-Dihydroxybenzoic acid (protocatechuic acid)	3.2	C_7_H_6_O_4_	154.0266	153.0193	109
3-Hydroxybenzoic acid	4.8	C_7_H_6_O_3_	138.0317	137.0244	93
3-Hydroxyphenylpropionic acid	4.9	C_9_H_10_O_3_	166.0630	165.0557	121
Dihydroxybenzoic acid	5.1	C_7_H_6_O_4_	154.0266	153.0193	109
3,4-Dihydroxyphenylpropionic acid	5.4	C_9_H_10_O_4_	182.0579	181.0506	137; 109
Hydroxyphenylacetic acid	5.4	C_8_H_8_O_3_	152.0473	151.0401	107
3-Methoxy-4-hydroxybenzoic acid (vanillic acid)	5.9	C_8_H_8_O_4_	168.0423	167.0350	152; 108
4-Hydroxybenzoic acid	6.1	C_7_H_6_O_3_	138.0317	137.0244	93
Dihydroxybenzoic acid	6.4	C_7_H_6_O_4_	154.0266	153.0193	109
3-Methoxy-4-hydroxyphenylacetic acid (Homovanillic acid)	6.4	C_9_H_10_O_4_	182.0579	181.0506	137; 122
Dihydroxybenzoic acid	6.7	C_7_H_6_O_4_	154.0266	153.0193	109
Methoxy-hydroxybenzoic acid	6.8	C_8_H_8_O_4_	168.0423	167.0350	108
Hydroxyphenylacetic acid	8.0	C_8_H_8_O_3_	152.0473	151.0401	107
Dihydroxybenzoic acid glucose	8.1	C_13_H_16_O_9_	316.0794	315.0722	153
3-Methoxy-4-hydroxyphenylpropionic acid	8.4	C_10_H_12_O_4_	196.0736	195.0663	136
4-Hydroxyphenylpropionic acid	8.7	C_9_H_10_O_3_	166.0630	165.0557	121
Methoxy-hydroxyphenylpropionic acid	9.8	C_10_H_12_O_4_	196.0736	195.0663	136
Methoxy-hydroxybenzoic acid	11.3	C_8_H_8_O_4_	168.0423	167.0350	152; 108

**Table 3 molecules-27-06510-t003:** Content of individual phenolic compounds present in *Brunfelsia grandiflora*. Results represent the mean ± standard deviation (*n* = 4). N.D.: not detected; d.w.: dry weight.

RT (min)	Proposed Compound	*Brunfelsia grandiflora* (mg/100 g d.w.)
**HYDROXYCINNAMIC ACIDS and HYDROXYCINNAMATES**		
4.7	5-Chlorogenic acid	3.13 ± 0.18
5.0	Caffeoylquinic acid	1.89 ± 0.07
5.1	Caffeoylquinic acid	0.70 ± 0.05
5.9	Caffeic acid	0.21 ± 0.01
8.0	*p*-Coumaric acid	0.10 ± 0.01
9.0	Ferulic acid	3.99 ± 0.10
9.4	Sinapic acid	0.31 ± 0.02
10.2	Dehydrodiferulic acid	7.62 ± 0.11
11.3	Caffeic acid-*O*-glucoside	533.86 ± 8.29
12.5	Coumaric acid-*O*-glucoside	57.57 ± 0.50
12.8	Coumaroylquinic acid	2.26 ± 0.10
13.0	Ferulic acid-*O*-glucoside	391.46 ± 17.08
13.1	Sinapic acid-*O*-glucoside	81.55 ± 1.66
13.4	Ferulic acid-*O*-glucoside	19.33 ± 1.00
13.4	Feruloylquinic acid	151.04 ± 4.07
13.5	Sinapoylquinic acid	6.54 ± 0.09
13.7	Feruloylquinic acid	7.51 ± 0.22
13.8	Ferulic acid-*O*-glucoside	64.22 ± 2.11
15.2	Sinapic acid-*O*-glucoside	8.76 ± 0.30
15.6	Coumaric acid-*O*-glucoside	1.61 ± 0.14
17.4	Coumaric acid-*O*-glucoside	1.48 ± 0.04
	*TOTAL HYDROXYCINNAMIC ACIDS (mg/100 g) (%)*	*1345.13 ± 36.16 (66.77%)*
**HYDROXYCOUMARINS**		
3.7	Esculin	4.71 ± 1.16
5.7	Esculetin	21.49 ± 0.66
8.9	Scopoletin	286.77 ± 21.28
	*TOTAL HYDROXYCOUMARINS (mg/100 g) (%)*	*312.97 ± 23.11 (15.13%)*
**LIGNANS**		
8.8	Pinoresinol	0.77 ± 0.03
9.5	Matairesinol	2.45 ± 0.29
9.6	Hydroxysecoisolariciresinol isomer	5.19 ± 0.19
9.8	Secoisolariciresinol isomer	3.35 ± 0.20
9.9	Hydroxysecoisolariciresinol isomer	4.43 ± 0.10
11.3	Sesamol	55.36 ± 2.46
11.8	Secoisolariciresinol	2.42 ± 0.09
12.7	Cyclolariciresinol or Isolariciresinol	9.70 ± 0.57
12.9	Hydroxymatairesinol/Nortrachelogenin	15.23 ± 0.75
13.4	Sesamin	16.00 ± 0.47
15.2	Secoisolariciresinol isomer	2.95 ± 0.20
19.4	Episesamin	5.51 ± 0.13
	*TOTAL LIGNANS (mg/100 g) (%)*	*123.36 ± 5.48 (6.12%)*
**FLAVONOLS**		
4.9	Kaempherol-*O*-rutinoside	3.97 ± 2.21
10.6	Isorhamnetin-*O*-rutinoside	8.09 ± 0.09
12.0	Kaempherol-*O*-galactoside-*O*-rhamnoside	7.62 ± 0.29
12.2	Isorhamnetin-*O*-glucoside-*O*-rhamnoside	94.51 ± 2.47
	*TOTAL FLAVONOLS (mg/100 g) (%)*	*114.18 ± 5.06 (5.67%)*
	**PHENOLIC ACIDS**	
2.3	Methoxy-hydroxybenzoic acid glucoside	2.86 ± 0.22
2.4	Dihydroxybenzoic acid glucose	2.16 ± 0.10
2.6	Dihydroxybenzoic acid glucose	2.80 ± 0.20
3.2	3,4-Dihydroxybenzoic acid (protocatechuic acid)	5.23 ± 0.25
4.8	3-Hydroxybenzoic acid	6.97 ± 0.15
4.9	3-Hydroxyphenylpropionic acid	3.99 ± 0.24
5.1	Dihydroxybenzoic acid	1.03 ± 0.04
5.4	3,4-Dihydroxyphenylpropionic acid	3.81 ± 0.15
5.9	3-Methoxy-4-hydroxybenzoic acid (vanillic acid)	2.73 ± 0.16
6.1	4-Hydroxybenzoic acid	1.50 ± 0.02
6.4	Dihydroxybenzoic acid	1.22 ± 0.06
6.4	3-Methoxy-4-hydroxyphenylacetic acid (Homovanillic acid)	0.17 ± 0.01
6.5	4-Hydroxyphenylacetic acid	7.61 ± 0.09
6.7	Dihydroxybenzoic acid	0.59 ± 0.06
6.8	Methoxy-hydroxybenzoic acid	2.11 ± 0.06
8.0	Hydroxyphenylacetic acid	3.98 ± 0.07
8.1	Dihydroxybenzoic acid glucose	0.81 ± 0.02
8.4	3-Methoxy-4-hydroxyphenylpropionic acid	1.08 ± 0.05
8.7	4-Hydroxyphenylpropionic acid	2.73 ± 0.10
9.8	Methoxy-hydroxyphenylpropionic acid	1.85 ± 0.07
11.3	Methoxy-hydroxybenzoic acid	7.23 ± 0.24
	*TOTAL PHENOLIC ACIDS (mg/100 g) (%)*	*62.46 ± 2.38 (3.10%)*
**GALLATES**		
2.0	Gallic acid	0.97 ± 0.13
3.6	Methyl-gallate	0.66 ± 0.06
5.0	Galloyl-glucose	1.82 ± 0.13
6.3	Ethyl-gallate	3.99 ± 0.09
6.6	Methyl-gallate	2.89 ± 0.10
7.2	Ethyl-gallate	12.38 ± 0.38
9.4	Methyl-gallate	0.57 ± 0.06
11.4	Ethyl-gallate	19.92 ± 0.56
13.0	Ethyl-gallate	3.28 ± 0.10
	TOTAL GALLATES (mg/100 g) (%)	*46.48 ± 1.62 (2.31%)*
**FLAVANOLS**		
6.1	Gallocatechin	5.54 ± 0.12
14.2	Methyl-epigallocatechin	0.29 ± 0.02
	TOTAL FLAVANOLS (mg/100 g) (%)	*5.83 ± 0.14 (0.29%)*
**FLAVANONES**		
10.2	Eriodictyol	0.54 ± 0.05
12.1	Naringenin-*O*-glucoside	1.59 ± 0.06
12.3	Eriodictyol-*O*-glucoside	0.65 ± 0.02
15.9	Naringenin	0.06 ± 0.01
16.2	Eriodictyol-*O*-glucoside	1.00 ± 0.07
16.9	Hesperetin	0.09 ± 0.01
18.0	Naringenin-*O*-glucoside	0.37 ± 0.05
	*TOTAL FLAVANONES (mg/100 g) (%)*	*4.30 ± 0.27 (0.21%)*
	**TOTAL PHENOLIC COMPOUNDS**	** *2014.71 ± 74.23 (100%)* **

## Data Availability

Not applicable.

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
