# Peer review of "Identification, Quantification, and Characterization of the Phenolic Fraction of Brunfelsia grandiflora: In Vitro Antioxidant Capacity"

_molecules, 2022, doi:10.3390/molecules27196510_

Round 1
Reviewer 1 Report
Manuscript: molecules-1946011
The authors of the manuscript entitled “Phenolic fraction of Brunfelsia grandiflora: identification, quantification and characterization of their antioxidant potency” present data obtained from the extraction of compounds from the bark of Brunfelsia grandiflora. Reported results contribute to the knowledge of compounds produced by this plant used in traditional medicine as a remedy against various illness symptoms, mainly about phenolic fraction involved in the antioxidant activity.
The information provided would be useful to the scientific community and therefore I think the manuscript deserves to be published in “Molecules”.
I have just a few suggestions:
1. Please check the title carefully.
2. Since the main sources of phenolic compounds are considered fruits and vegetables, in the “Introduction” or the “Conclusion” sections the authors should briefly introduce the bark as an important source of polyphenols, especially in relation to the large availability of this part of the plants as wood waste which could provide a high amount of therapeutic compounds.
3. The paper “A Critical Review of Phenolic Compounds Extracted from the Bark of Woody Vascular Plants and Their Potential Biological Activity” Molecules 2019, 24, 1182;doi:10.3390/molecules24061182”, should be added to the Reference list and used for the discussion.
4. The “Reference” section should be implemented by more recent articles on Brunfelsia grandiflora. As example: “Luzuriaga-Quichimbo CX, Hernández Del Barco M, Blanco-Salas J, Cerón-Martínez CE, Ruiz-Téllez T. Chiricaspi (Brunfelsia grandiflora, Solanaceae), a Pharmacologically Promising Plant. Plants (Basel). 2018 Aug 18;7(3):67. doi: 10.3390/plants7030067”. Furthermore, more recent scientific publications on research on the pharmacological action of B. grandiflora need to be added. As example: “A New Leishmanicidal Saponin from Brunfelsia grandiflora. Hiroyuki Fuchino, Setsuko Sekita, Kanami Mori, Nobuo Kawahara, Motoyoshi Satake, Fumiyuki Kiuchi. Chemical and Pharmaceutical Bulletin 2008 Volume 56 Issue 1 Pages 93-96”.
5. Line 98: (inhibitory concentration 50 μg/mL) is not the meaning of IC50 that is correctly reported after. Please correct.
Author Response
Response to Reviewer 1 Comments
The authors would like to thank the referees for their revisions. We feel that their comments have greatly contributed to improving our work. The changes made in the manuscript have been shaded in yellow (referee 1) and green (referee 2).
The authors of the manuscript entitled “Phenolic fraction of Brunfelsia grandiflora: identification, quantification and characterization of their antioxidant potency” present data obtained from the extraction of compounds from the bark of Brunfelsia grandiflora. Reported results contribute to the knowledge of compounds produced by this plant used in traditional medicine as a remedy against various illness symptoms, mainly about phenolic fraction involved in the antioxidant activity. The information provided would be useful to the scientific community and therefore I think the manuscript deserves to be published in “Molecules”.
I have just a few suggestions:
- Please check the title carefully.
Since the proposed title misleads the study’s objective, it is proposed to replace the title with the following one that better reflects the results presented: Identification, quantification and characterization of the phenolic fraction of Brunfelsia grandiflora. In vitro antioxidant capacity.
- Since the main sources of phenolic compounds are considered fruits and vegetables, in the “Introduction” or the “Conclusion” sections the authors should briefly introduce the bark as an important source of polyphenols, especially in relation to the large availability of this part of the plants as wood waste which could provide a high amount of therapeutic compounds.
Thank you for your comment. The bark was introduced in the introduction and at the beginning of the Results and Discussion section, as has been suggested by the reviewer.
- The paper “A Critical Review of Phenolic Compounds Extracted from the Bark of Woody Vascular Plants and Their Potential Biological Activity” Molecules 2019, 24, 1182;doi:10.3390/molecules24061182”, should be added to the Reference list and used for the discussion.
Thank you for your right suggestion. This paper was cited in the Results and Discussion section and added to the reference list.
- The “Reference” section should be implemented by more recent articles on Brunfelsia grandiflora. As example: “Luzuriaga-Quichimbo CX, Hernández Del Barco M, Blanco-Salas J, Cerón-Martínez CE, Ruiz-Téllez T. Chiricaspi (Brunfelsia grandiflora, Solanaceae), a Pharmacologically Promising Plant. Plants (Basel). 2018 Aug 18;7(3):67. doi: 10.3390/plants7030067”. Furthermore, more recent scientific publications on research on the pharmacological action of B. grandiflora need to be added. As example: “A New Leishmanicidal Saponin from Brunfelsia grandiflora. Hiroyuki Fuchino, Setsuko Sekita, Kanami Mori, Nobuo Kawahara, Motoyoshi Satake, Fumiyuki Kiuchi. Chemical and Pharmaceutical Bulletin 2008 Volume 56 Issue 1 Pages 93-96”.
These references were included, briefly disussed in the results and discussion section and cited in the reference list. Thank you for your suggestion.
- Line 98: (inhibitory concentration 50 μg/mL) is not the meaning of IC50 that is correctly reported after. Please correct.
The mistake was corrected, thank you.

Reviewer 2 Report
I am very grateful you for the invitation to review the manuscript molecules-1946011 by Mateos and coauthors "Phenolic fraction of Brunfelsia grandiflora: identification, quantification and characterization of their antioxidant potency". The aim of this study was to identify the phenolic composition of this medicinal plant to know the chemical structures of these phytochemicals that are behind the renown biological properties of Brunfelsia grandiflora. The work is interesting but needs adjustments to increase the quality of the material.
Comments:
- Abstract, Lines 18-19: Include the associated proven medicinal properties.
- Abstract: Briefly include the steps for identifying the compounds (obtaining, extracting and other steps).
- Abstract, Line 27: Change “hides” to “evidence” or another similar word.
- Abstract, Lines 28-29: In the sentence “supports its therapeutic properties used by our ancestors”, indicates only therapeutic properties scientifically proven.
- Keywords, Lines 30-31: Change the repeated keywords by different words from the title.
- Introduction: Specify production characteristics and feasibility of use.
- Material and methods: I suggest the inclusion of an image of “Brumbelsia grandiflora” to illustrate the botanical specifications highlighted in the introduction (Lines: 44-52)
- Material and methods, Line 297: Specify if there was a standardization of granulometry.
- Results, Lines 77-79: The authors should highlight only health effects that have already been scientifically proven. Popular reports should be associated with the clinical determinations of molecules.
- Results, Line 96: Insert the difference between the methods and what type of component each of them evaluates.
- Table 1: Standardize the textual source used in Table 1.
- Results: Indicate generally how different groups can act or enhance the antioxidant effect.
- The results are well presented and clearly identify the components. Insert correlation mentioned earlier.
Author Response
Response to Reviewer 2 Comments
The authors would like to thank the referees for their revisions. We feel that their comments have greatly contributed to improving our work. The changes made in the manuscript have been shaded in yellow (referee 1) and green (referee 2).
I am very grateful you for the invitation to review the manuscript molecules-1946011 by Mateos and coauthors "Phenolic fraction of Brunfelsia grandiflora: identification, quantification and characterization of their antioxidant potency". The aim of this study was to identify the phenolic composition of this medicinal plant to know the chemical structures of these phytochemicals that are behind the renown biological properties of Brunfelsia grandiflora. The work is interesting but needs adjustments to increase the quality of the material.
Comments:
- Abstract, Lines 18-19: Include the associated proven medicinal properties.
Thank you for your suggestion. Proven medicinal properties were included in the abstract.
- Abstract: Briefly include the steps for identifying the compounds (obtaining, extracting and other steps).
Steps followed to isolate polyphenols from the bark of the Brunfelsia grandiflora previous to their chemical characterization were included in the abstract.
- Abstract, Line 27: Change “hides” to “evidence” or another similar word.
‘Hides’ was changed by ‘evidences’ as the reviewer has suggested.
- Abstract, Lines 28-29: In the sentence “supports its therapeutic properties used by our ancestors”, indicates only therapeutic properties scientifically proven.
‘Supports is therapeutic properties used by our ancestors’ was changed by ‘Supports its therapeutic properties scientifically proven’ as the reviewer has suggested.
- Keywords, Lines 30-31: Change the repeated keywords by different words from the title.
The repeated keywords by different words from the title were changed, although Brunfelsia grandiflora was maintained due to its importance in this study.
- Introduction: Specify production characteristics and feasibility of use.
To our knowledge, Brunfelsia grandiflora species known as "Chiric Sanango" is sold in the medicinal plant markets, especially in the Amazon regions and in the capital Lima, from wild populations or home gardens, similar to other medicinal species, and its trade is based on extractivism. This idea was included in the introduction.
- Material and methods: I suggest the inclusion of an image of “Brumbelsia grandiflora” to illustrate the botanical specifications highlighted in the introduction (Lines: 44-52)
An image of Brunfelsia grandiflora was included in the introduction.
- Material and methods, Line 297: Specify if there was a standardization of granulometry.
The standardization of granulometry was not carried out, we are sorry.
- Results, Lines 77-79: The authors should highlight only health effects that have already been scientifically proven. Popular reports should be associated with the clinical determinations of molecules.
We understand the reviewer's point of view, but along those lines, we only wanted to insist on our ancestors' acceptance of this plant based on its potential health benefits. To clarify this aspect, the following tagline will be added at the end of the sentence 'although no scientific studies confirm most of these effects’.
- Results, Line 96: Insert the difference between the methods and what type of component each of them evaluates.
As the differences between ABTS and DPPH are detailed in material and methods section, in the old line 96 is briefly explained to avoid redundancies.
- Table 1: Standardize the textual source used in Table 1.
Textual source used in Table 1 was standardize.
- Results: Indicate generally how different groups can act or enhance the antioxidant effect.
A brief structure-antioxidant activity of the identified compounds is included in the Results and Discussion section.
- The results are well presented and clearly identify the components. Insert correlation mentioned earlier.
The correlations were inserted.